# Thermal Comfort Assessment in Urban Green Spaces: Contribution of Thermography to the Study of Thermal Variation between Tree Canopies and Air Temperature

Alexandre Ornelas [1,4,*], António Cordeiro [2,4] and José Miguel Lameiras [3,5]

1 Institute of Interdisciplinary Research (IIIUC), University of Coimbra, 3030-789 Coimbra, Portugal
2 Faculty of Arts and Humanities, University of Coimbra, 3004-504 Coimbra, Portugal; rochettecordeiro@fl.uc.pt
3 Faculty of Sciences, University of Porto, 4169-007 Porto, Portugal; jmlameiras@fc.up.pt
4 Centre for Interdisciplinary Studies (CEIS20), 3000-186 Coimbra, Portugal
5 BIOPOLIS/CIBIO—Research Centre in Biodiversity and Genetic Resources, 4485-661 Vairão, Portugal
* Correspondence: alexandre.ornelas@uc.pt

**Abstract:** Understanding the thermal effects of different urban patterns that constitute today's urban landscapes is critical to the development of urban resilience to climate change. This article aims to assess the efficiency of urban green spaces in thermal regulation. Through thermography, we explored the interaction between air temperature and the spatial components within these environments. Through comparative analysis involving a UAV, we studied the relationship between air temperatures at varying altitudes and the temperature within tree canopies. The results revealed significant differences in the thermal distribution between impervious urban areas with buildings and green spaces. These findings provide important information for assessing thermal comfort and the efficiency of urban green spaces in mitigating the impact of extreme heat events. During the summer months, green spaces, due to shade and the enhanced absorption of solar radiation by trees, exhibited lower temperatures compared to impervious areas. However, in winter, urban areas displayed higher temperatures, attributable to their heat retention capacity. This study contributes to the existing knowledge base by providing an in-depth examination of the thermal efficiency of urban green spaces across different layers of their lower atmosphere. Our results underscore the crucial role of tree cover in thermal comfort regulation, offering valuable information for sustainable urban planning. These insights are particularly relevant for the design of more comfortable and resilient environments in response to climatic variations and for the crafting of a tree-planting strategy in Mediterranean climate cities, an area where the impacts of climate change are becoming increasingly apparent.

**Keywords:** thermal comfort; urban green spaces; urban tree canopy; urban tree cooling effect; thermography; UAV; thermal efficiency; air temperature; Coimbra





## 1. Introduction

At a time when the planet is in need for an extraordinary global effort to mitigate the negative effects of climate change, it is in the densely urbanised and populated urban areas that action is imperative [1]. Even though cities cover only a small fraction of the Earth (<2% of the Earth's surface), more than 50% of the world's population lives in urban areas, and this number is increasing [2]. Land use changes in the urban surfaces (edification and surface imperviousness) as well as industry, traffic, and the general activities of the cities are the driving forces behind the increase in the sources of carbon dioxide emissions. These can be assumed to be the main drivers of global climate change [3,4].

In urban areas, as natural environments are gradually replaced by man-made impervious surfaces, different urban morphologies and environments are being created. There is a need to understand these new urban ecosystems and their potential climate effect in an

inclusive and integrated manner. As described by the authors of [5], cities are to be understood as complex adaptive systems composed of interconnected socio-economic-political systems, ecological–biophysical-climate systems, and technological infrastructure systems. As Lévi-Strauss said, the city "is the most complex of human inventions . . . at the confluence between nature and artificial" [6,7]. In this context, it can be understood that these interdependent systems have dynamic properties which are often unpredictable, non-linear, and emerging [8,9]. The uncertainty of these processes and challenges regarding urban planning, politics, and urban ecology have the potential to improve people's well-being and urban habitability itself [10–14]. This is aligned with the prospect of sustainable cities (and territories), constituted by a society aware of its role as a transformative agent of the space [15,16].

In this context, it is not surprising that the phenomenon of "urban heat islands" (UHIs), which characterises district heating, in comparison with the surrounding rural environment, has established itself as key a indicator of the urban climate [3,17–21]. As an easy phenomenon to define and measure, this is the most studied of the effects of urban climate change. The mitigation of the effects of UHI through the spatial configuration of green spaces in the sustainable design of urban environments has thus become a matter of growing concern under climate change [3,22–35]; these have been researched as supporting principles for the design and planning of Nature-Based Solutions (NBS) as mitigation strategies towards the negative effects of Climate Change in Urban Areas [36–38].

There are several reasons why it is important to understand the temperatures at these different levels. Firstly, tree canopies play a crucial role in the thermal regulation of urban areas [39–43]. On the other hand, assessing temperatures at heights above the canopies allows for an understanding of heat dissipation and air circulation in the permit area. This is particularly important for assessing the influence of UTC on urban ventilation and the dissipation of accumulated heat [44–47]. The presence of trees helps to reduce daytime temperatures by absorbing solar radiation, providing shade, and increasing evapotranspiration through the cooling effect of urban trees [48–50]. Aerial thermography with UAVs allows for the rapid analysis of temperature distribution in green spaces and the effect of vegetation cover on thermal regulation. Thermographic images show temperature variations, particularly in the tree canopies, and highlight the cooling effect of vegetation [51–53].

## 2. Materials and Methods

### 2.1. Definition of the Study Area

The city of Coimbra, Portugal, is located at medium latitude (40° North), has about 100,000 inhabitants, and has a series of green spaces that are very well defined within the historic centre. The climate is Mediterranean with a significant oceanic influence (it is located only 30 km from the coast, with an average annual rainfall of only about 956.6 mm, concentrated mainly in the coldest period), reflecting important microclimatic nuances that are mainly due to a very specific morphology; its eastern sector is characterised by a small mountain that is about 500 metres in height, which presents the central sector dominated by flattened levels at 100/120 metres separated by small but sharp valleys, the sector where the city has developed over two millennia, while its western sector is characterised by a wide alluvial plain open to the ocean air masses and low slopes [54]. This complex morphological context makes Coimbra a good case study for a pilot project on urban and microtopoclimatology in the Mediterranean belt, where heat waves (and cold waves) have been increasing over the last two decades.

The selection of the study area was carried out according to previous studies on the city of Coimbra, such as those conducted by the authors of [55,56], and more recently the study conducted by the authors of [57], which identified the "Urban Heat Islands" in consolidated urban areas. The urban context of the university campus (Polo I) located in the historic area of the city, where a UHI was identified, as well as the presence of a botanical garden, a contiguous green space within which different green structures mosaic

(Figure 1), provides an interesting case study that contributes to the development of more sustainable planning strategies.

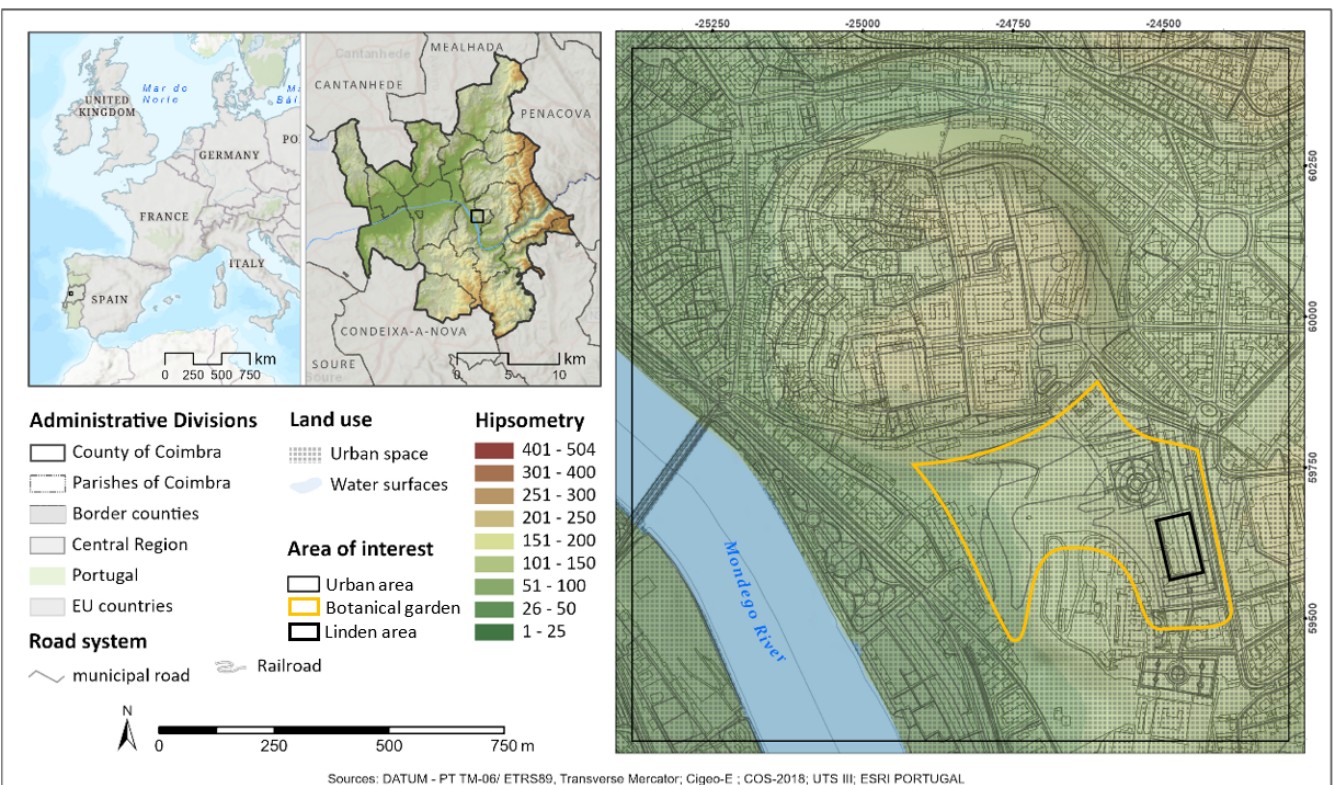

**Figure 1.** Framework of the study area.

This study analyses the significance of a 13-hectare urban green space, the Botanical Garden of the University of Coimbra (JB), which holds great importance for the city.

The study area features a corridor of linden trees (32 m in height); together with its surrounding area, this provided an opportunity to obtain comparative data on different green structure patterns and densities, as well as open impervious spaces.

It should also be noted that, due to the topography of Coimbra, the botanical garden is in a sheltered area, and our measurements revealed that it is distant enough from the river to the point where it does not cause significant thermal influence. This possible influence of the river is not manifested neither at 1.5 m above the ground nor at the tree canopy level. In the surveyed period, the dominant wind came from the east (from the botanical garden towards the direction of the river) and was very mild.

### 2.2. Research Methods

A methodology involving the use of a mobile temperature sensor and a UAV was used to study surface and altitude temperatures in the Botanical Garden of the University of Coimbra. Since the aim was always to work with real values of temperature, it was necessary to combine different techniques for the collection of data.

The aim of this work was to study the temperature variations during the day, in the morning, and in the afternoon in two different seasons: summer and winter. We set out to analyse the temporal, spatial, and seasonal temperature variations at the surface and at altitude to better understand the thermal behaviour of the environment and its relationship with climatic and seasonal factors.

The different temperature layers (air temperature, ground temperature, and canopy temperature) play a very important role in our global analysis. In this paper, we wanted to focus on the temperature values observed at the upper "surface" of the tree canopy,

commonly referred to as the Urban Tree Canopy (UTC), as well as the different layers within the canopy and the layer developed just above the canopy.

We attempted to establish a relationship between 2D spatialization with a component of temperature data collection in a pedestrian way and the understanding of thermal profiles at altitude. This served to provide a 3D understanding as well as the temperatures found in trees, on the ground, among others. In this way, a model that can interact and react to the changes that occur was obtained and used to compare the temperatures at altitude, in the objects through thermography, and at the surface at 1.5 m from the ground. Several scales of analysis that contribute to a better understanding of the thermal behaviour of an arboreal sector of a single species well were collected.

By analysing the temperatures in these layers, the impact of UTC on mitigating the urban heat island effect can be assessed. Thus, by considering the different temperature stratums, we can obtain a slightly more comprehensive and detailed view of urban thermal patterns, understand the role of UTC in thermal regulation, and subsequently identify areas with potential for improvements in sustainability and environmental comfort in urban areas.

The database is the result of combining 2 campaigns carried out on 8 July 2022 and 31 January 2023, with the collection of temperatures, and consequently the statistical treatment of the data and the preparation of thematic cartography, being carried out in the morning (9:30 a.m.) and in the afternoon (3:30 p.m.). In particular, the campaigns took place in an area populated by a group of trees of the species *Tilia x europaea*, commonly known as linden, which allowed us to study a site of relevant importance because it is a deciduous species. In winter, this species, with its fallen leaves, allows solar radiation to penetrate, while in summer, with its developed foliage, it creates areas of shade. Therefore, this space is a good example to draw some conclusions about bioclimatic comfort and its application in urban planning.

As an example, in the case of the pedestrian paths (Figure 2A), several equally calibrated data loggers (Tinytag Plus 2—TGP-4020, Chichester, West Sussex, UK) were used, and temperature data were collected with temperature sensors placed 1.5 m above the ground to measure the ambient temperature, i.e., the temperature felt by the citizens [58]. Another issue considered was direct sunlight, so precautions were taken to avoid exposing the probes to direct sunlight. At each of the points of the paths (always with common points between them), the time of arrival and the permanence of the same was recorded for about one minute in order to stabilise the sensor. The data were then processed by performing a statistical analysis which involved calculating the range of values during the time interval in which the probe was stabilised, thus obtaining the average temperature for that location. Then, in order to carry out the thematic mapping of the temperature with the points collected and their temperature values, together with their respective coordinates, we obtained a map for each moment of the day using ArcGIS Pro through a tool called Empirical Bayesian Kriging.

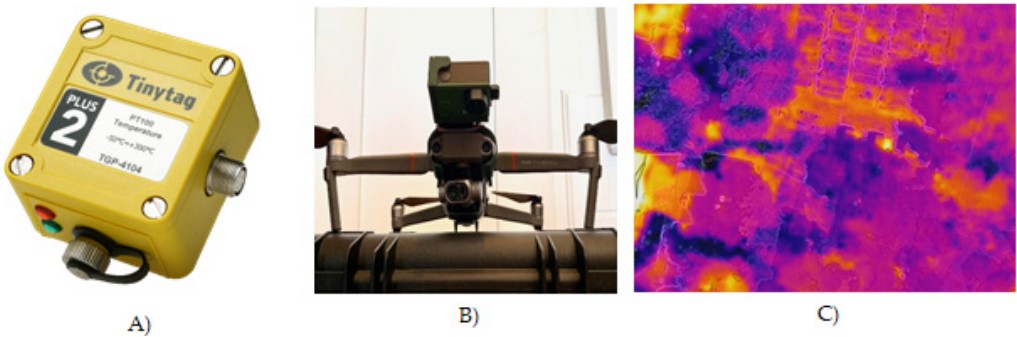

**Figure 2.** (**A**) Tinytag Plus 2—TGP-4020; (**B**) DJI Mavic Enterprise 2; (**C**) Sample of sensor area view and vertical measurement of air temperature.

The data were then processed, and a statistical analysis was carried out, which consisted of calculating the range of values during the time interval in which the probe was already stabilised and then taking the average (a method also used in aerial surveys). The points collected, together with their temperature values and corresponding coordinates, were used to produce the thematic temperature map. Using ArcGIS Pro and the Empirical Bayesian Kriging tool [58], it was possible to obtain a temperature map for each moment of the day.

For collecting vertical temperature data (Figure 2B), a highly accurate temperature sensor (Tinytag 2 Plus data logger) was attached to the drone. This sensor continuously gathered temperature readings at various heights as real-time recordings were conducted second by second, generating vertical profiles at altitude during a continuous descent at a speed of 1 m/s. To ensure minimal interference from the rotor vortex, the sensor was securely positioned on a cable situated 10 m below the drone.

By adopting this approach, comprehensive temperature data were obtained throughout the vertical range, offering valuable insights into the thermal profiles of the lower atmosphere as the drone descended. The methodology employed in this study closely followed the description provided by the authors of [58], ensuring methodological consistency and reliable results.

The mapping method used was similar to that of the pedestrian paths, except for the fact that, in this work, where the altitude variable was entered, Empirical Bayesian Kriging 3D was used as described by the authors of [58].

To obtain the temperatures by thermography (Figure 2C), the same drone, the DJI Mavic Enterprise 2, was used, and this drone also contains a thermal camera (M2ED Thermal Camera) with an uncooled VOx microbolometer sensor, lens HFOV: 57°, aperture: f/1. 1, sensor resolution $160 \times 120$, pixel pitch 12 μm, spectral band 8–14 μm, accuracy high gain: max ±5% (typical) and low gain: max ±10% (typical), scene range: high gain: −10 °C to +140 °C and low gain: −10 °C to +400 °C. To measure temperatures on the surfaces of objects, the flight altitude was set at 75 m to ensure adequate coverage and good spatial resolution without affecting the collected values [51]. By monitoring the changes in temperature using thermography, we can visualise the variations in different sectors of the plants over time, gathering information on their thermal regulation strategies and adaptability.

By processing the data collected by the thermal camera, through videos and photographs in the visible and thermal bands, it was possible to map the different temperatures [51]. This mapping consisted of identifying the temperature at each location, considering the very high, very low, and medium temperature points. Each point obtained had a pair of coordinates and an associated temperature value. Based on these points, a network of points was created, resulting in a temperature map for each moment of the day. The Empirical Bayesian Kriging method previously explained for the footpath was applied. This procedure was applied to all the data collected, both in the thermal images and in the visible images, at different times of the day and in the different seasons of the year. In this way, it was possible to obtain detailed cartographic representations of the temperatures recorded, allowing for a comprehensive spatial analysis of the data collected.

### 2.3. Analysis of Air Temperature in the Surrounding Area

To provide context for the temperatures occurring in the space around JB, a number of maps are presented that show the interactions between the green and urbanised spaces. Two well-defined (Figure 3), distinct green areas are found, the JB (A) and the Jardim da Sereia—JS (B), separated by a distance of 290 m, and surrounding them are the different types of buildings with varying heights and sizes as well as streets with different types of pavement, such as cobblestones and asphalt. Our research approach was also influenced by topography and sun exposure.

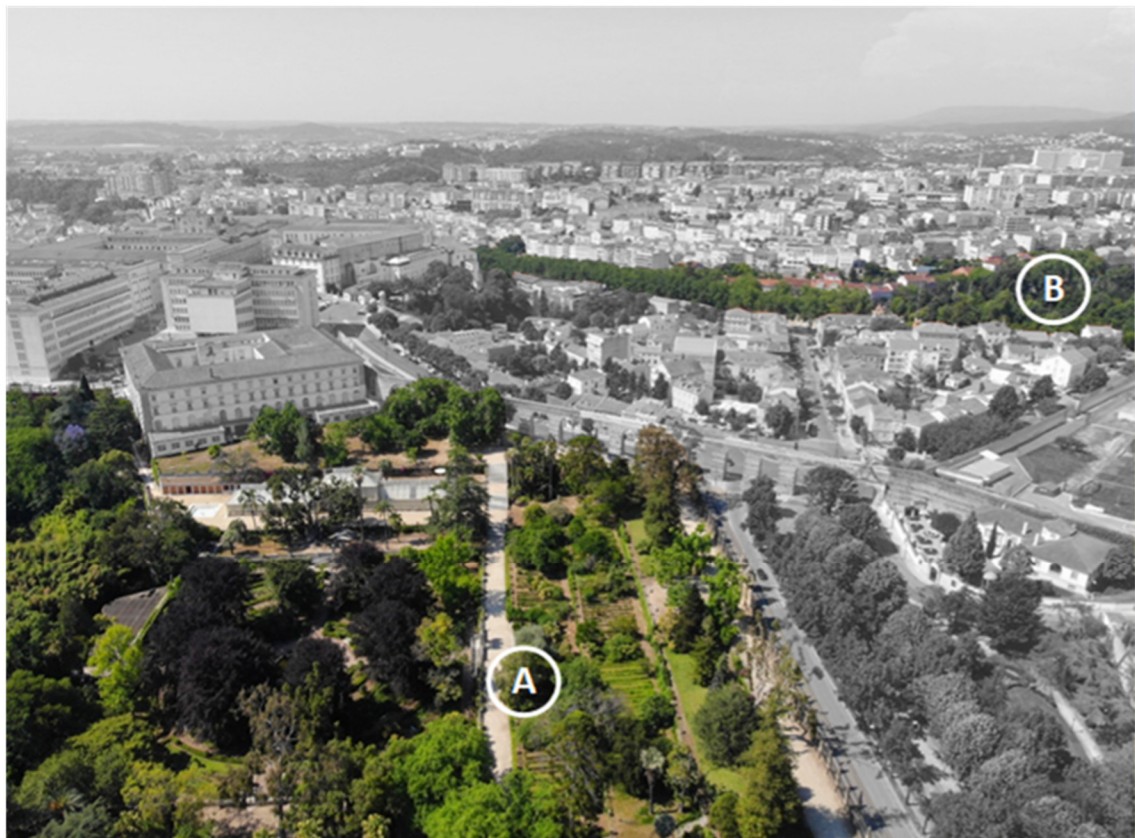

**Figure 3.** Visualisation of green spaces and urbanised spaces.

In Figure 4, we present the temperatures at different times of the day and in different seasons in such a way in order to highlight the main areas of interest. The aim is to contextualise the importance of green spaces in the city of Coimbra. We have chosen to keep an irregular shape to highlight the temperature points collected on the walking paths, avoiding introducing too much noise in this analysis. In summer, in the morning (Figure 4A), we observed a difference of 6.44 °C between the maximum (33.38 °C) and the minimum (26.95 °C) recorded temperature. In JB, we found cooler temperature values (between 27 °C and 28 °C) when analysing the surrounding area. In the afternoon, the maximum temperature was 42.02 °C and the minimum was 34.75 °C, resulting in a temperature amplitude variation of 7.27 °C. Again, it can be observed that JB and JS have the lowest temperatures during this period. In the afternoon, the difference between the thermal amplitudes is more significant, but in the green areas, it is possible to find a more pleasant thermal sensation due to the lower temperatures. In places without green areas, it is possible to observe the formation of 5 distinct urban heat islands which occur in dense urban areas with high solar radiation. The temperatures recorded in these places exceed 40 °C.

Regarding the winter analysis, in the morning period, the maximum temperature ($T_{máx}$) was 7.80 °C and the minimum temperature ($T_{min}$) was 2.43 °C, giving an amplitude of 5.36 °C due to the accumulation of radiation. Again, JB and JS are the locations with the lowest temperatures. In the afternoon, the $T_{máx}$ value was 22.64 °C, and the $T_{min}$ was 21.13 °C, resulting in a variation of only 1.51 °C. This reduced variation is a favourable indication in terms of bioclimatic comfort, especially considering the cold winter context due to the radiation received.

In this case, during the winter, the green areas become the coldest places in the study area. However, in the afternoon, based on a sample of trees in JB and JS, a smaller amplitude of temperature variation is observed. This observation contradicts the notion that, in winter, we find areas with uninviting temperatures within the green spaces. This fact can be

attributed to the presence of different species of trees, many of them deciduous and others perennial, which contribute to more efficient thermal regulation and less temperature variation.

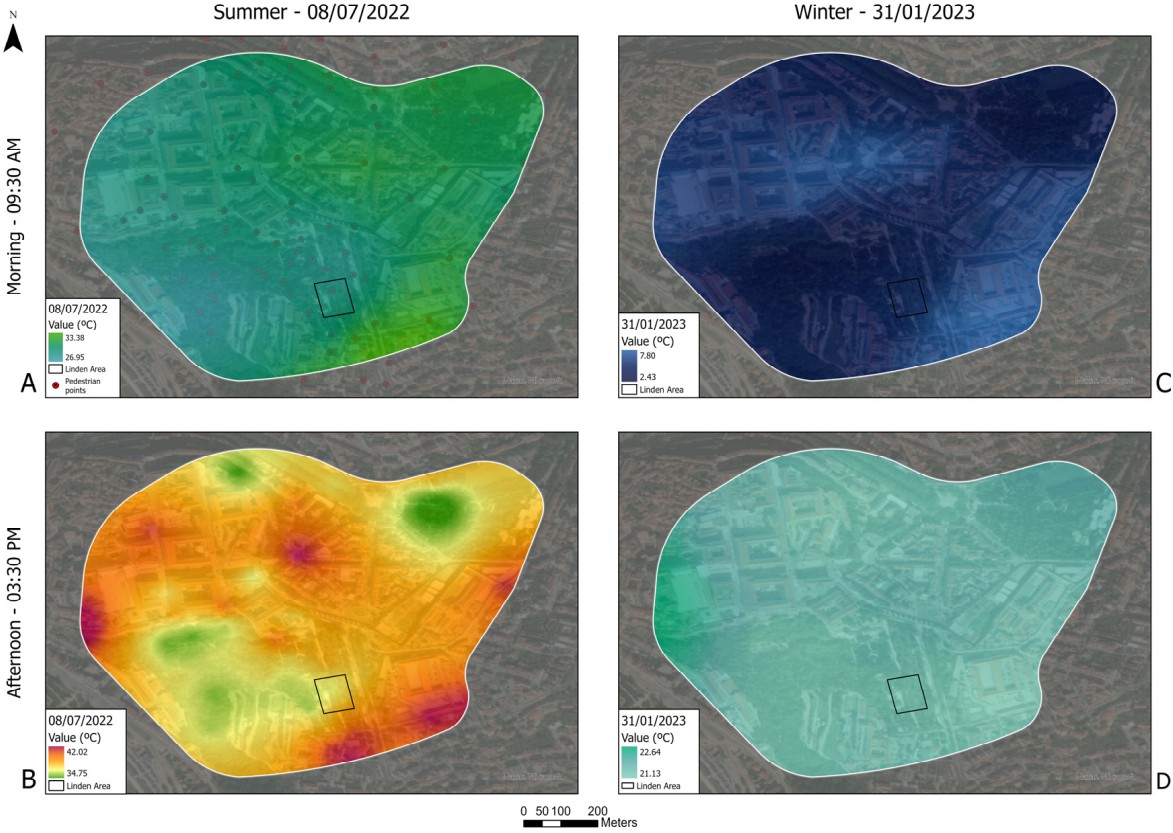

**Figure 4.** Air temperature in the different seasons of the year.

Based on this general analysis, a more detailed presentation of the research results are presented in Section 3, with a specific focus on the linden corridor.

## 3. Results

### 3.1. Surface Temperatures at the City of Coimbra

#### 3.1.1. Analysis of Summer Temperatures—8 July 2022

The linden corridor offers an intriguing analytical context distinct from others. When the different layers of information are compared, it can be observed that the interactions between air temperature and object temperature render this site a relevant case study for the local micro-climate. It is relevant to juxtapose an image in the visible band with another in the thermal band. At this stage, the analysis scale is presented without temperature values, with blue signifying colder areas and red indicating warmer areas. This research approach centred around understanding the impact of a connected green structure, specifically a green corridor, and the systemic effects of a group of trees rather than focusing on the effects of individual trees. The analysis was conducted collectively to assess the overall influence of these green spaces. The reason behind this decision lies in the fact that the tree canopies blend together to form an interconnected structure, making it challenging to isolate and define the contribution of each crown separately. However, this approach enables us to draw meaningful conclusions regarding the systemic effects of the tree groups as a whole.

In the morning (Figure 5), it can be observed that the homogeneity of the colours represented reflects a situation of equilibrium; the temperatures in this period are very similar. The areas with less vegetation are those that receive more solar radiation, which

results in warmer colours in the thermal images. The shadows are not as pronounced, but it can be seen that they correspond to places with lower temperatures.

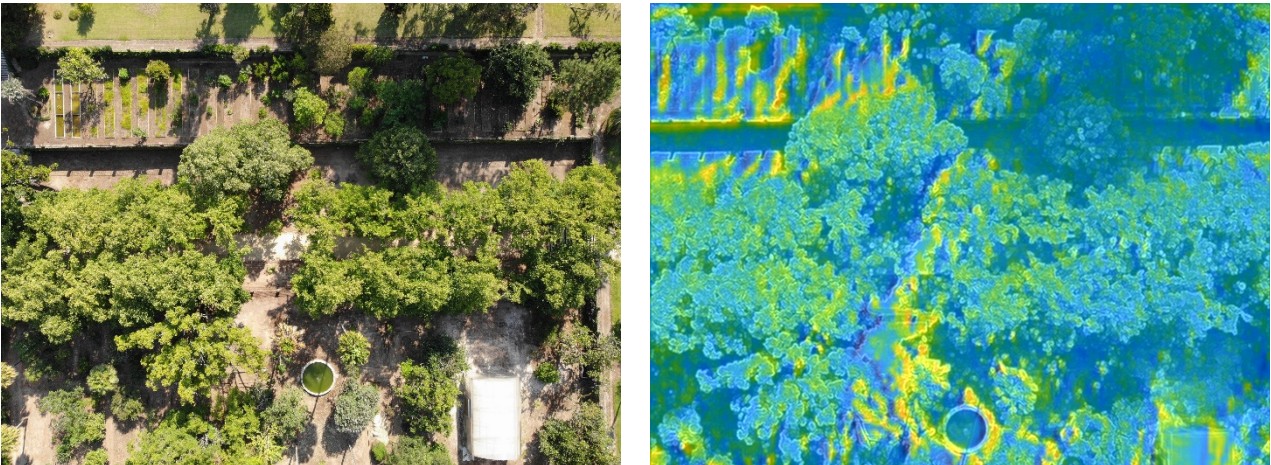

**Figure 5.** Morning photo 8 July 2022—09:30 a.m. (**left**: RGB; **right**: thermal).

In the afternoon (Figure 6), asymmetries are clearly visible, with greater heterogeneity in areas with less vegetation, warm colours in areas without shade, and lower temperatures in areas with shade. In the tree canopy, on the other hand, the colour is largely uniform, indicating that the temperature values are similar in the central area, where the linden trees are found, and outside the central corridor, where other tree species are found. The close relationship between the maximums and minimums in this image shows the importance of the tree canopy, which protects the maximum solar radiation and maintains the sense of coolness.

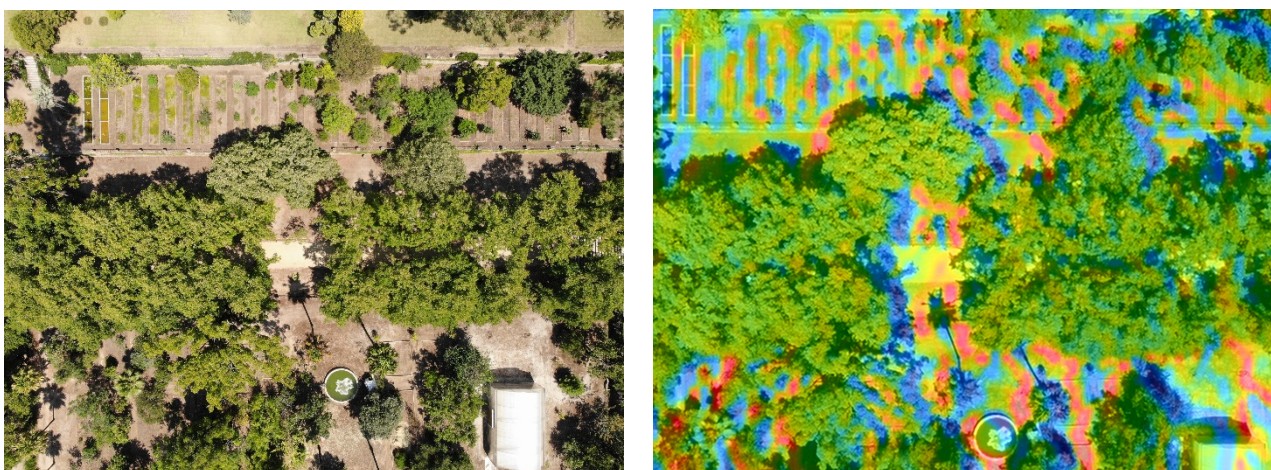

**Figure 6.** Afternoon photo 8 July 2022—3:30 p.m. (**left**: RGB; **right**: thermal).

To summarise the information presented in Figure 7, we have organised the data so that the morning representation is on the left of the figure and the afternoon representation on the right. The area we are analysed is 6079.68 m$^2$ (0.61 ha). The upper part shows the temperature of the objects (thermography), as explained in the methodology. In this case, we have used a single colour scale, where the coldest temperatures are represented in blue (since we only visualise the coldest temperatures on 31 January 2023), and the warmest temperatures are represented in the transition from red to purple.

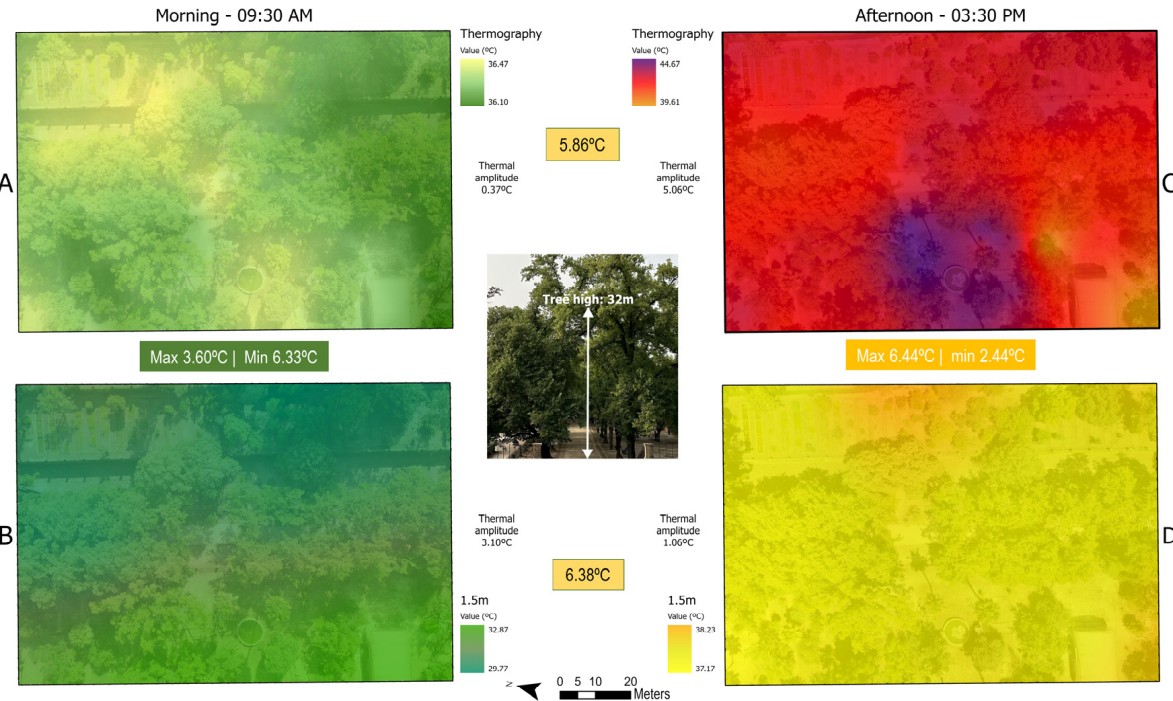

**Figure 7.** Comparison of temperatures on 8 July 2022.

During the morning, looking at the thermographic map (Figure 7A), we can observe an image in which the $T_{máx}$ was 36.47 °C and the $T_{min}$ was 36.10 °C. The thermal amplitude of 0.37 °C is a not very significant value, which indicates that the solar radiation in the early morning is not so strong, and we could not find significant differences between the temperatures in the canopy and on the ground.

After analysing the image representing the temperatures at 1.5 m from the ground, it was found that the $T_{máx}$ was 32.87 °C, and the $T_{min}$ was 29.77 °C. In the western part of the image (Figure 7B), the temperatures are higher than in the eastern part, and this is due to solar exposure. In the east, in this area and time, the solar exposure does not have the capacity to heat all sectors, especially those in the shade, resulting in lower temperatures. In this case, the temperature variation is significant, with a temperature range of 3.10 °C.

Between Figure 7A,B we can see a rectangle representing the thermal amplitude between the $T_{máx}$ and $T_{min}$ of each figure. That is, in this rectangle, the Max represents the difference between the $T_{máx}$ of Figure 7A and the $T_{máx}$ of Figure 7B, which has a thermal amplitude value of 3.60 °C. On the other hand, the Min represents the difference between the $T_{min}$ of Figure 7A and the $T_{min}$ of Figure 7B, with a thermal amplitude variation of 5.33 °C. In the case of the Max, in certain areas and in the tree canopy, the temperature is 3.60 °C higher than that recorded at the lower level (1.5 m from the ground). Furthermore, there are places where the ground temperature is higher than the temperature above it, at a height of 1.5 m. On the other hand, after analysing the Min, we found that the amplitude is greater compared to the Max, and there are places where the temperature on the ground is 6.33 °C higher than the temperature recorded in the air at 1.5 m height.

In the linden corridor, the temperature values show greater variation. In the tree canopy, the temperature varies between 36.15 °C and 36.38 °C, while below the tree canopy, at a height of 1.5 m, the temperature varies between 30.35 °C and 31.51 °C, with a maximum amplitude of 4.87 °C and a minimum of 5.8 °C. It is in the lower temperature values that we notice the importance of the shade provided by the trees, and they are able to provide this protection due to their ability to retain some of the radiation and prevent it from reaching the ground.

In the afternoon, thanks to our thermographic analysis (Figure 7C), we can verify that the $T_{máx}$ was 44.67 °C and that the $T_{min}$ was 39.61 °C. Due to it being such a small space,

we can confirm that, in areas without vegetation, the ground temperature is much higher than areas with vegetation, resulting in a temperature range of 5.06 °C. In the tree canopy in the central zone of the lime trees, temperatures range between 41.69 °C and 42.73 °C, while on the ground, we recorded the highest values in this image.

Regarding the analysis at 1.5 m (Figure 7D), the thermal amplitude is smaller and presents a value of 1.06 °C, and the $T_{máx}$ is 38.23 °C; the $T_{min}$ is 37.17 °C. Due to the reduced air circulation, the oscillation of the values is minimal, which shows that the thermal differences are in equilibrium.

Therefore, as in the previous analysis for the morning, we follow the same method to address the rectangle between Figure 7C,D. In this case, the Max has a value of 6.44 °C, which indicates a very high temperature difference between the tree canopy and the ground, higher than that the air temperature at 1.5 m (height). On the other hand, the Min has a value of 2.44 °C, reflecting a smaller amplitude between the $T_{min}$ of the two figures being compared. Such a value shows a smaller difference between the temperatures in the objects and the air temperature, indicating a greater thermal balance. After comparing the temperature in the tree canopy (42.93 °C) with the temperature felt under the tree canopy, at a height of 1.5 m, we observed that the temperature varies between 37.34 °C and 37.89 °C. In the opening between the linden trees, the temperature felt on the ground is 42.88 °C, and on the ground near the water fountain, the temperature felt is the absolute maximum, 44.67 °C.

### 3.1.2. Analysis of Winter Temperatures—31 January 2023

The following analysis is similar to the one described earlier but contains minor variations. On 31 January 2023 (Figure 8), it can be seen that the deciduous trees have shed their leaves and that the linden trees are devoid of any leaves. It can be observed that the solar incidence is greater, indicating that the trees offer almost no shade, except for their trunks. In the analysis of the thermal band, it can be seen that the areas represented in red exhibit higher temperatures due to direct radiation and that there are multiple places with elevated temperatures. The spectrum of colours is broader, suggesting that there will be potential thermal variations once the collected points are analysed.

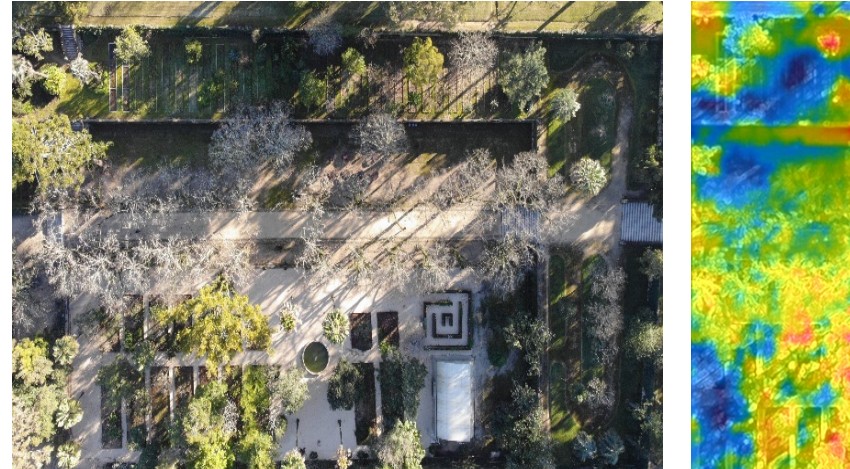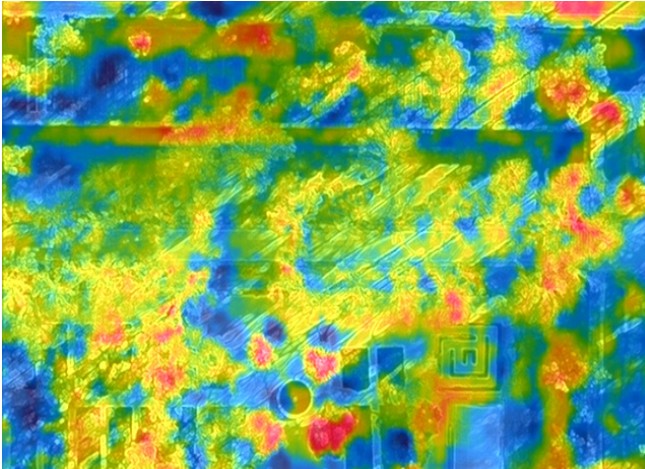

**Figure 8.** Morning photo 31 January 2023—09:30 a.m. (**left**: RGB; **right**: thermal).

During the afternoon (Figure 9), as the sun's position changes, it was observed that the intensity of sunlight on the ground diminishes. The evergreen trees, being exposed to the sun's radiation, create shaded zones, which are prevalent in the area. Furthermore, it can be confirmed that the presence of a nearby hill and the lower position of the sun at this time of year result in a larger shadow area visible in the image. In the thermal band, it can be seen that the incoming radiation at the tops of the trees and within the tree canopy leads

to a temperature increase. The absence of leaves also results in a portion of the ground being heated.

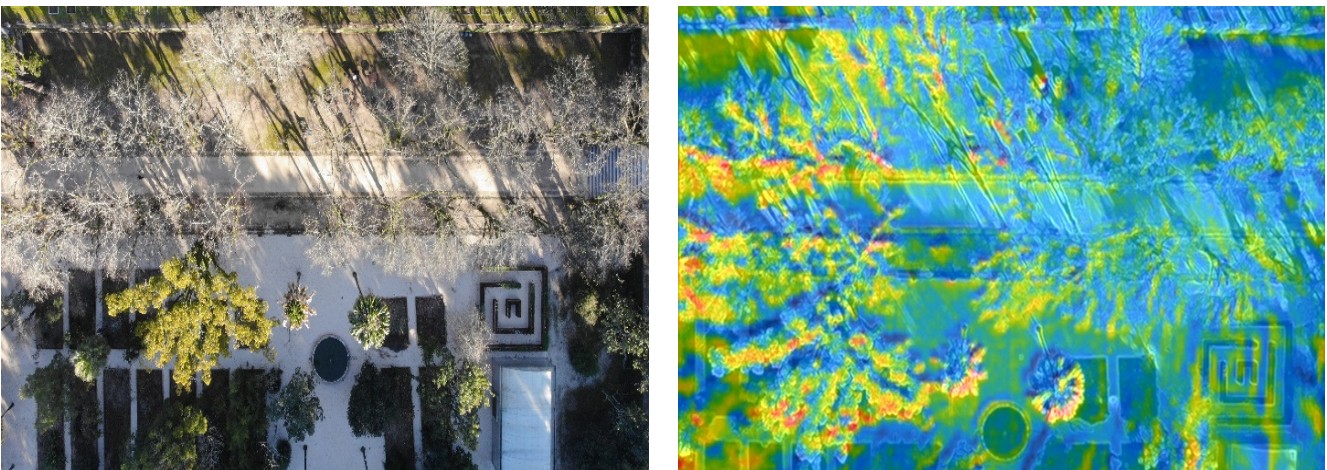

**Figure 9.** Afternoon photo 31 January 2023—3:30 p.m. (**left**: RGB; **right**: thermal).

The changes that occur when comparing the temperature in the objects and in the air will be analysed next. In the morning of this day, on the thermographic map (Figure 10A), an image can be observed where the $T_{máx}$ was 6.55 °C and the $T_{min}$ was 5.13 °C. The temperature range was 1.42 °C and, as it was early in the morning, the solar radiation did not have enough time to heat all the objects at this location. However, this difference was found to be negligible.

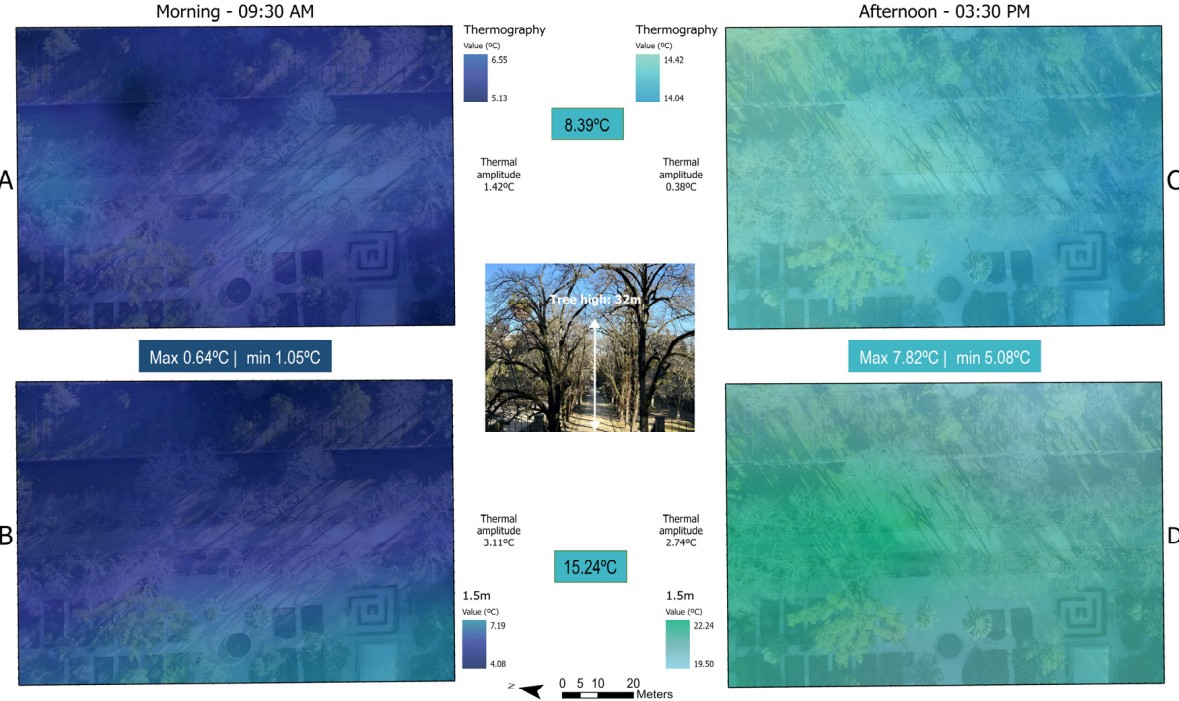

**Figure 10.** Comparison of temperatures on 31 January 2023.

In the case of the temperature at 1.5 m (Figure 10B), after examination, it was found that the $T_{máx}$ was 7.19 °C, and the $T_{min}$ was 4.08 °C, resulting in a thermal amplitude of 3.11 °C. This means that, in some locations, the presence of light radiation sufficiently to warms the air temperature, while in locations more isolated from light, very low temperatures are found to be more difficult to warm.

The Max for this location is 0.64 °C, which represents a very straightforward balance at the level of thermal variation; there is no great heating capacity at this location during this period. After analysing the Min, we were able to verify that the amplitude was 1.05 °C, which indicates a greater ease of heating the objects in relation to the air. In the locations where the temperature is lower in Figure 10A compared to Figure 10B, we can observe that it becomes more difficult to heat the air temperature in the locations where the temperature is equally low.

When the focus was shifted to the afternoon period in the temperature obtained by thermography (Figure 10C), it was found that the $T_{máx}$ was 14.42 °C and the $T_{min}$ was 14.04 °C. Again, the thermal amplitude was reduced, with the temperature varying by 0.38 °C, proving that the asymmetries are minimal at this location. On the other hand, the air temperature was observed to have warmed compared to the morning, in which the $T_{máx}$ was 22.24 °C and the $T_{min}$ was 19.50 °C, and the thermal amplitude in this interval was 2.74 °C. Specifically, the widest range of warmer temperatures is shown by the corridor of linden trees. Without foliage, temperatures ranged from approximately 21.92 °C to 20.95 °C, while in other areas, the temperatures were lower.

After analysing the variation in the Max temperature value, we observed that the air temperature is 7.82 °C higher than in the objects captured by thermography. The reason for this large difference is the way in which different materials with different thermal conductivities heat up. In the case of Min, the recorded value is 5.08 °C, and although the objects are cold, the air manages to heat up and create a pleasant thermal sensation.

## 4. Discussion

These results can be analysed at different scales, and reflection can occur on summer mornings. When a green area (Figure 7B) is compared with an urban area (Figure 4A), it can be observed that the temperatures are slightly lower in the green area. However, it is important to point out that, in this comparison, there are moments when the solar radiation shines more on the green areas than on some of the buildings due to the different heights that create shaded areas. Therefore, the temperature tends to be very similar between them.

During the afternoon, the situation is completely different; after a few hours of sun exposure, the air temperature in urban spaces (Figure 4B) reaches approximately ($\approx$) 42 °C, which is significantly higher than the temperature we recorded for a green space with well-defined tree cover (Figure 7D), wherein the temperature was $\approx$37.50 °C. These green spaces provide bioclimatic comfort, providing cooler temperatures and contributing to the mitigation of extreme heat climatic events.

In winter, the temperature distribution is different. During a winter morning, in episodes of extreme cold, urban spaces (Figure 4C) manage to maintain a higher temperature, around 7 °C, while in green spaces (Figure 10B), the temperature is $\approx$5 °C. However, in our research area, we observed a greater oscillation, with a temperature range of 3.11 °C. This means that, in some locations, the warmer temperatures recorded outside JB can also be found in spaces similar to JB. On a winter afternoon, the temperatures tend to equalise. In the urban space (Figure 4D), the temperature is $\approx$22 °C, and in the green space (Figure 10D), the temperature is $\approx$22 °C. However, there are exceptions in our research area where temperatures of $\approx$20 °C have been recorded. In the context of bioclimatic comfort, we can feel this variation of 2 °C both in the morning and in the afternoon.

In order to compare the morning and the afternoon periods, we have summarised the relevant information in Table 1, which shows a comparison of the two temperatures analysed in the lime tree research area. Some of these values have already been presented in Figures 7 and 10, but here, they are organised in a more easily digestible way.

**Table 1.** Comparative analysis of temperatures (°C) in the different seasons.

| Summer—8 July 2022 | | | | | | | |
|---|---|---|---|---|---|---|---|
| Thermography | 9:30 a.m. | 3:30 p.m. | Difference (p.m.–a.m.) | 1.5 m | 9:30 a.m. | 3:30 p.m. | Difference (p.m.–a.m.) |
| $T_{máx}$ | 36.47 | 44.67 | 8.2 | $T_{máx}$ | 32.87 | 38.23 | 5.36 |
| $T_{min}$ | 36.1 | 39.61 | 3.51 | $T_{min}$ | 29.77 | 37.17 | 7.4 |
| Thermal amplitude $(T_{máx} - T_{min})$ | 0.37 | 5.06 | NA | Thermal amplitude $(T_{máx} - T_{min})$ | 3.1 | 1.06 | NA |
| Mean $(T_{máx} - T_{min})$ | 36.29 | 42.14 | **5.85** | Mean $(T_{máx} - T_{min})$ | 31.32 | 37.7 | **6.38** |
| Winter—31 January 2023 | | | | | | | |
| Thermography | 9:30 a.m. | 3:30 p.m. | Difference (p.m.–a.m.) | 1.5 m | 9:30 a.m. | 3:30 p.m. | Difference (p.m.–a.m.) |
| $T_{máx}$ | 6.55 | 14.42 | 7.87 | $T_{máx}$ | 7.19 | 22.24 | 15.05 |
| $T_{min}$ | 5.13 | 14.04 | 8.91 | $T_{min}$ | 4.08 | 19.5 | 15.42 |
| Thermal amplitude $(T_{máx} - T_{min})$ | 1.42 | 0.38 | NA | Thermal amplitude $(T_{máx} - T_{min})$ | 3.11 | 2.74 | NA |
| Mean $(T_{máx} - T_{min})$ | 5.84 | 14.23 | **8.39** | Mean $(T_{máx} - T_{min})$ | 5.635 | 20.87 | **15.24** |

In terms of thermography, in summer, from morning to afternoon, we observed a heating of 8.2 °C in the $T_{máx}$ of the objects. On the other hand, in the air temperature at 1.5 m, this heating was 5.36 °C. Regarding the $T_{min}$, there was a warming of 3.51 °C from the morning to the afternoon in the thermography, and at 1.5 m, the warming of the air was 7.4 °C. So, we can say that the $T_{máx}$ warms objects the most, while in the 1.5 m measurements, the $T_{min}$ is the one that warms the most throughout the day. By analysing the mean $(T_{máx} + T_{min})$ for these periods, we can see that, in terms of thermography, the average increase in temperature from morning to afternoon was 5.86 °C, and the average increase in temperature at 1.5 m was 6.38 °C. In winter, when we analyse the $T_{máx}$ in terms of thermography, we can see that there is an increase of 7.87 °C in the temperature of the objects, while at 1.5 m, the temperature in the air increases by 15.05 °C. At $T_{min}$, the thermography values increased by 8.91 °C, and at 1.5 m, the increase was 15.42 °C from morning to afternoon. In summary, the thermography shows an average increase in object temperature of 8.39 °C, while the air temperature shows an average increase of 15.24 °C. These values are consistent with the differences identified earlier regarding temperature increase from morning to afternoon.

To provide a visual spatialisation of the different temperature values obtained from the thermography and the air temperature measurements, we have tried to organise these data according to a scheme (Figure 11). This composite analysis includes the air temperatures obtained by the UAV at different heights (from 0 m to 50 m) using a thermal sensor, as well as the air temperature at 1.5 m from the ground measured using the same sensor. In addition, the temperature values in the canopy trees and on the ground in the unshaded areas are also shown. Concerning the visual scale, the trees are ≈32 m high (as mentioned above), the children are less than 1.5 m, and the height of the adults varies between 1.60 m and 1.85 m.

In the thermal profile obtained by the UAV, the following temperatures were recorded. On the ground, a temperature of 37.87 °C was recorded; immediately after the sensor leaves the ground, the temperature is 37.88 °C. Up to a height of seven metres, the temperature increases to 37.97 °C; from this height, the temperature begins to decrease, reaching 37.36 °C at 32.6 m. At 50 m, the temperature is 36.84 °C. The temperature at 1.5 m from the ground is 37.89 °C, and at the top of the tree canopy, the temperature is 42.73 °C. Finally, the highest temperature recorded in this analysis at ground level in full sunlight was 42.88 °C. Indirectly, we know that, in a shaded situation, the temperature on the ground was 37.87 °C.

By comparing the air temperature (37.36 °C) at 32.6 m with the canopy temperature (42.73 °C) at 32 m, we noticed that there is a significant difference. The thermal amplitude recorded is 5.37 °C, which reflects the interaction between trees and solar radiation and evidences the ability of the trees to absorb, reflect, and filter solar radiation. From an ecophysiological perspective, *Tilia x europaea* opens its leaf stomata during the day as an adaptive way to promote evapotranspiration and better control water and temperature.

This leads to a slight increase in the humidity of the air, which explains the higher thermal inertia that was measured.

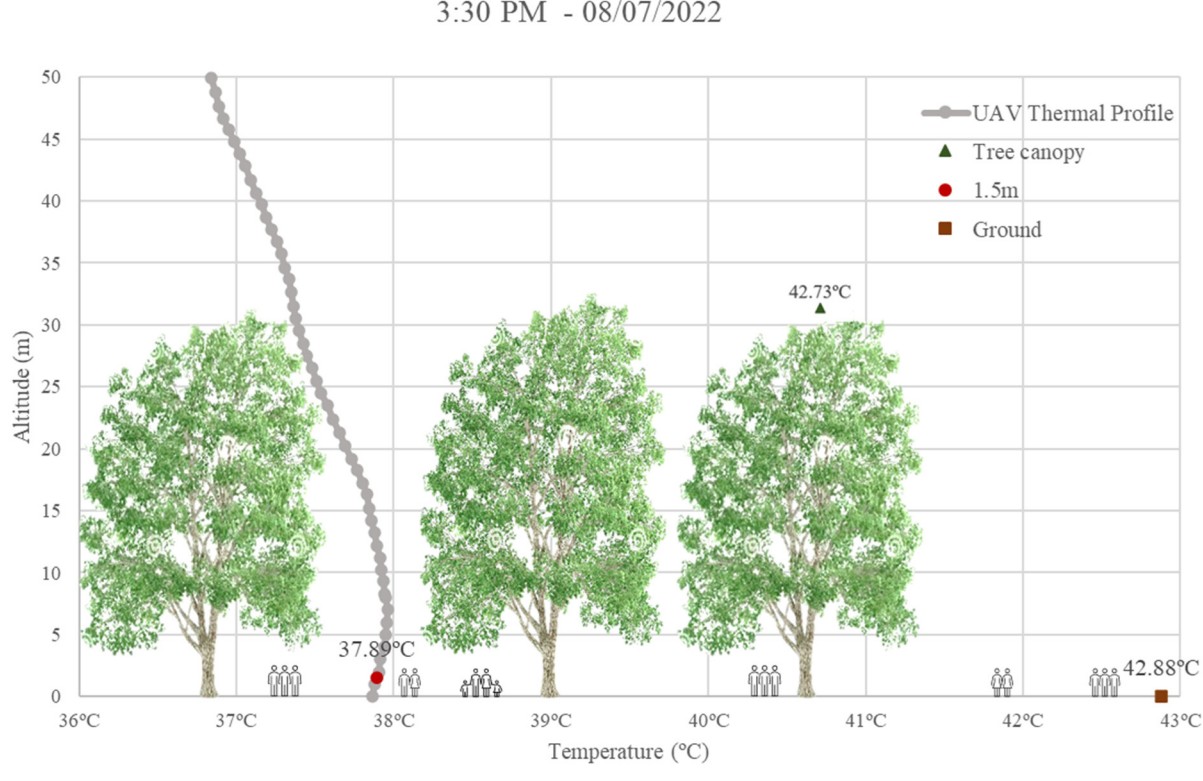

**Figure 11.** Final scheme of the spatialisation of the different temperatures.

For a comparison with the previous analysis, see Figure 12, which shows the thermal profile required at altitude in a treeless area to avoid the influence of irrelevant factors caused by the environment. We can see that the profile is completely different from the one shown previously in Figure 11.

Our thermal profile analysis (Figure 12) revealed the following temperatures: the air temperature on the ground was 39.06 °C; just after the sensor leaves the ground, the temperature was 38.95 °C. The temperature recorded at a height of 1.5 m from the ground was 38.45 °C. Up to a height of 5.3 m, the temperature drops to 37.77 °C; from this height, the temperature begins to fall more sharply, reaching 35.75 °C at 14.8 m, and from this point, the temperature continues to decrease until it reaches 50 m with a temperature of 35.06 °C.

This study contributes to and enriches the knowledge developed in previous studies on the thermal efficiency of urban green spaces. It is distinguished by its use of real temperature values and complements analyses available in the literature by also considering the temperatures present in objects such as trees and the ground. The thermal regulation provided by a few tree densities in the afternoon can be compared with the results reported by the authors of [49,50]. This is complemented by the role of evapotranspiration in thermal inertia, which explains not only the shading effect of trees by intercepting solar radiation but also the reduction in energy absorption and storage by urban surfaces [51]. These effects have been identified and demonstrated in this thermal profile (UAV Thermal Profile), which reflects the cooling potential of urban trees from the surface level. This shows that urban green spaces are generally cooler at the surface than more paved areas with a reduced presence of trees, an effect called Park Cool Island (PCI).

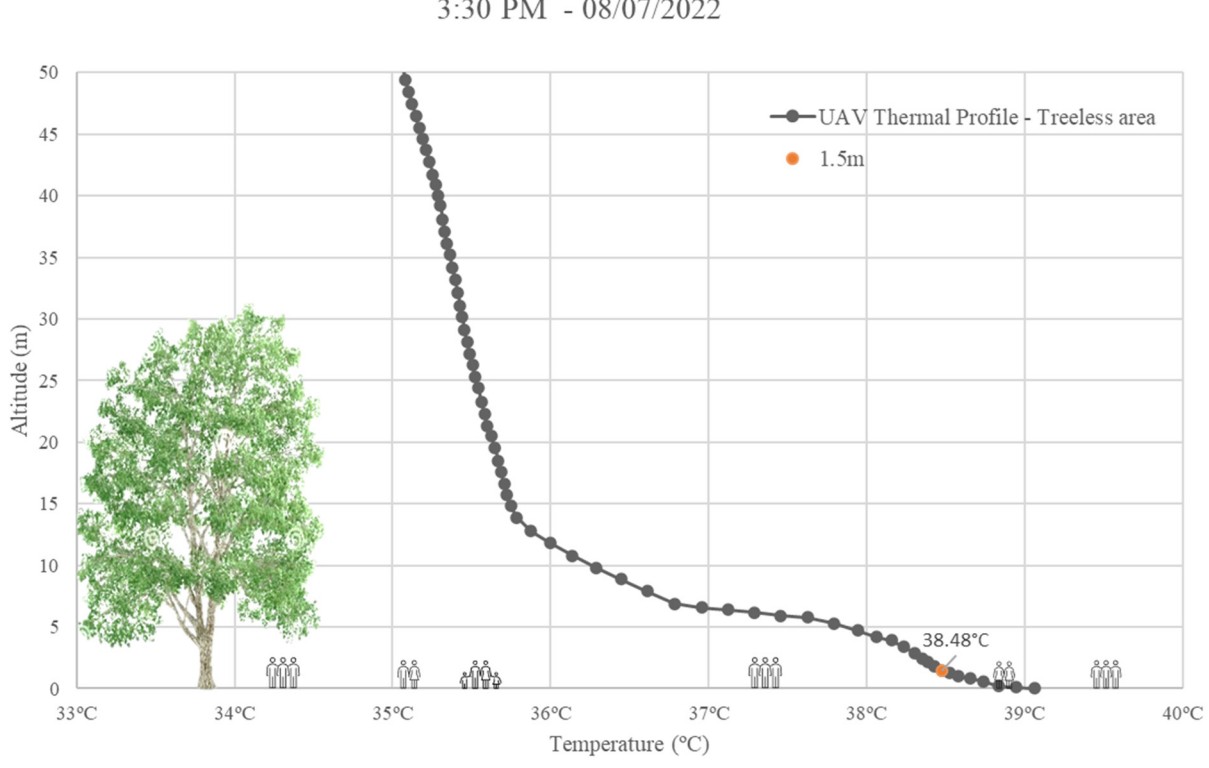

**Figure 12.** Treeless area of the spatialization of the different temperatures.

## 5. Conclusions

In this work, the thermal efficiency of urban green spaces and their role in mitigating the effects of urban climate have been investigated. Valuable information about the thermal distribution and heating patterns in different areas was gathered through air temperature analysis and the use of thermography. This research enables the identification of areas that offer greater efficiency during periods of intense heat and locations that provide thermal comfort in winter.

Green spaces have slightly lower temperatures than urban spaces, especially during the summer months. These spaces provide bioclimatic comfort, offering cooler temperatures and contributing to the climate resilience of urban areas.

Through aerial surveys and pedestrian paths, we were able to spatialise and map the thermal variations in different locations. It was important for us to incorporate real temperature data into our research, and thermography allowed us to obtain the temperatures of objects, complementing the analysis of air temperatures for each day analysed. A new correlation was found after verifying that, in the areas under study, there was an increase in temperature during the transition from morning to afternoon. It was found that the air temperature at 1.5 m above the ground showed a greater average temperature increase than the temperature of the objects, both in summer and in winter.

Many have recognised the fact that trees play a crucial role in regulating temperature by absorbing, reflecting, and filtering solar radiation. The distance between trees has also been noted as being important, allowing for air circulation and thus aiding decreases in temperature [42,47]. It has been found that the presence of trees triggers significant temperature changes in the air and in objects such as the ground. In summer, it was observed that the temperature in the tree canopy could exceed the surrounding air temperature, indicating the ability of trees to accumulate heat. The shade provided by the canopy of the linden trees was observed to reduce the air temperature by 4.84 °C at a height of 1.5 m, which was considered highly significant. In a paper by M. A. Rahman (2017), also in an area with linden trees, the cooling effect of trees (ΔTAir) was found to be about 3.5 °C [39].

The insights provided by this study should be used to guide the design of new urban green spaces and the redesign of existing ones. This climate-resilient strategy is closely aligned with the goals pertaining to the implementation of Nature-Based Solutions (NBS) and the provided ecosystem services. This strategy involves integrating intelligent microclimate design to promote urban resilience to the extreme weather events that have been occurring more frequently. With this in mind, cities should be designed (or redesigned) in a way that ensures bioclimatic comfort and benefits the well-being of their inhabitants, even amid constant climatic changes.

For this work, we integrated different data collection methodologies, combining the simultaneous acquisition of temperatures at altitude and tree canopy temperatures using thermography with a UAV with temperatures at 1.5 m in a large urban green space.

This study's scope is currently being applied to other green spaces in Coimbra, facilitating an in-depth and comprehensive analysis of the thermal effects of urban green spaces with varying sizes and spatial configurations. The methods utilised in Coimbra are also undergoing testing in other regions, notably in the city of Porto, with potential applicability to diverse climate zones. Subsequent research endeavours will delve into tree species and their individual contributions, providing valuable insights into how specific trees influence the thermal dynamics of urban environments. This holistic approach will contribute significantly to our understanding of the impact of urban green spaces on the microclimate and open new avenues for sustainable urban planning and environmental design.

The case studies presented in this research act as a living lab whose findings can play a vital role in future tree planting strategies.

**Author Contributions:** Conceptualization, A.O. and A.C.; methodology, A.O.; software, A.O.; validation, A.O., A.C. and J.M.L.; formal analysis, A.O., A.C. and J.M.L.; investigation, A.O., A.C. and J.M.L.; resources, A.O., A.C. and J.M.L.; data curation, A.O., A.C. and J.M.L.; writing—original draft preparation, A.O.; writing—review and editing, A.O., A.C. and J.M.L.; visualization, A.O., A.C. and J.M.L.; supervision, A.O., A.C. and J.M.L.; project administration, A.O., A.C. and J.M.L.; funding acquisition, A.O., A.C. and J.M.L. All authors have read and agreed to the published version of the manuscript.

**Funding:** This research was funded by national funds through the FCT—Fundação para a Ciência 456 e a Tecnologia, which were granted to I.P. through project UIDB/00460/2020.

**Data Availability Statement:** Not applicable.

**Conflicts of Interest:** The authors declare no conflict of interest.

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
