# Peer review of "Thermal Comfort Assessment in Urban Green Spaces: Contribution of Thermography to the Study of Thermal Variation between Tree Canopies and Air Temperature"

_land, doi:10.3390/land12081568_

Round 1

Reviewer 1 Report

In this manuscript, the mechanism of canopy in thermal regulation was presented via thermal imaging drone. The results show that trees play a crucial role in filtering solar radiation to regulate temperature, but these issues discussed in this paper have been confirmed by many related papers. It is interesting that the UAV thermal imaging technology was introduced, and the air temperature was investigated into different levels. Yet, there are some problems the authors need to concern.
1) Line 103: This paper selects the Botanical Garden of the University of Coimbra as the research area. There is a river on the west side of the area and a valley on the east side. It is well known that the lake-land breeze circulation has a great impact on the thermal environment. Please explain whether it affects the results ?

2) Line 276: What is the calculation method for the temperature distribution data at the height of 1.5 m, could you please clarify it? How is about the accuracy? Please provide more details here.

3) Line 425: In the longitudinal analysis, how does the thermal imaging data obtained by the UAV convert into air temperature at different heights of 0-50 m? Is the data reliable?

4) Line 431: To improve the reliability of conclusion, I suggest that authors provide a comparative experiment for the air temperature measurement at different heights from 0 to 50 m in the treeless area, for avoiding the influences of irrelevant factors caused by the environment.

Proofreading is recommended.

Author Response

Response to Reviewer 1 Comments

Point 1: Line 103: This paper selects the Botanical Garden of the University of Coimbra as the research area. There is a river on the west side of the area and a valley on the east side. It is well known that the lake-land breeze circulation has a great impact on the thermal environment. Please explain whether it affects the results?

Response 1:

This is a very interesting question; this research team has compiled more than 5 years of research data on the thermal behaviour of the city of Coimbra. This team is currency developing a research paper that focuses on the influence of the topography on the thermal behaviour Coimbra. In it was possible to verify that the thermal influence of the river does not reach the botanical garden. The dominant wind in the surveyed period comes from the east (from the botanic garden to the direction of the river) and was very mild.

The botanical garden is in a sheltered area, due to the topography of Coimbra, and is distant enough from the river so that its influence is not measured. The possible influence of the river is not manifested neither at 1.5 m above the ground nor at the tree canopy level.

This important explanation has been added to the research paper.

Point 2: Line 276: What is the calculation method for the temperature distribution data at the height of 1.5 m, could you please clarify it? How is about the accuracy? Please provide more details here.

Response 2:

Thanks for pointing this out, as the methods used were not clear enough in its original writing.

To better explain the adopted methodological approach, the writing of the methodology has been improved. Further details on the methods have been added in text.

The temperatures at 1,5m are not calculated, they are measured with highly accurate temperature sensor (Tinytag 2 Plus data logger). The accuracy of the temperature sensors is calibrated annually by a private company. After each temperature data collection, all values are normalised and adjusted to ensure that measurements are uniform and consistent across all analysed sites. These rigorous procedures are key to ensuring reliable and accurate results when analysing the thermal distribution in the surveyed areas.

Temperatures measured through thermography are either object (trees for example) or surface temperatures. The combination the methods is one of the original contributions of this research, as it allowed for a richer and in depth analysis of the thermal behaviour of different spatial patterns in urban green spaces.

Point 3: Line 425: In the longitudinal analysis, how does the thermal imaging data obtained by the UAV convert into air temperature at different heights of 0-50 m? Is the data reliable?

Response 3:

Same as before, the writing of the methodology used has been improved. The description and justification of the methods used has been revised.

The thermal image was not used to calculate the temperature at altitude. Vertical temperature data was measured using a dedicated, highly accurate temperature sensor (Tinytag 2 Plus data logger) attached to the drone, which continuously collects the temperatures at different heights, covering 0-50 metres. The sensor is positioned on a cable attached to the drone, 10m below it, the distance required to avoid interference from the rotor vortex. This method has already been tested and validated before. References have been included in the article.

Point 4: Line 431: To improve the reliability of conclusion, I suggest that authors provide a comparative experiment for the air temperature measurement at different heights from 0 to 50 m in the treeless area, for avoiding the influences of irrelevant factors caused by the environment.

Response 4:

This is an important point. Authors have added a new figure to the article, of the survey conducted in a treeless area. This provides a comparison tool with the other areas. Figure 12 shows the thermal profile at altitude in a treeless area, it is possible to verify that the thermal behaviour is different from the tree canopy areas (Figure 11).

Reviewer 2 Report

This paper is very interesting. Due to global warming, it is now necessary to know the effect of trees in urban planning.

However, this question should be addressed in this research:

1. Why is no data collected on crown diameter? Crown diameter is one of the characteristics of cooling effect in an area.

Otherwise the manuscript is well presented

Author Response

Response to Reviewer 2 Comments

This paper is very interesting. Due to global warming, it is now necessary to know the effect of trees in urban planning. Otherwise the manuscript is well presented

Point 1: Why is no data collected on crown diameter? Crown diameter is one of the characteristics of cooling effect in an area.

Response 1:

Thank you for the positive comment.

In this case, the research approach focused on the effect of connected green structure, a green corridor, and the systemic effect of trees. Trees were analysed collectively. Due to the characteristics of this area, we did not analyse the influence that the crown of each tree provides in terms of shade, because the canopies of the trees merge to form a green corridor, making it difficult to identify and define each crown in isolation. But allows for conclusion in the systemic effect of the groups of trees.

We included this explanation in the text to better help the reader understand the logic of behind the analysis of a green structure of trees.

Future work that is currently being developed by this team will explore this perspective and the individual contributions, for that we really appreciate this suggestion.

Reviewer 3 Report

 The specific original study is a contribution to the body of knowledge with a detailed analysis of the thermal efficiency of urban green spaces. The results obtained highlight the importance of trees in regulating thermal comfort. The present work provides important information for sustainable urban planning to create more comfortable and resilient environments facing climatic crisis.

This study is very well-conducted and structured and presents scientific soundness. Many thanks for the invitation to review this manuscript.

Besides, it can add to the advance of the scientific thematic field.

In this regard, I recommend the acceptance. The only comments I want to referre are some extra proposals related to other future research case studies presented other treespecies findings in different areas and results. Collecting all the results and combine them as a vital role in future tree planting strategies.

Author Response

Response to Reviewer 3 Comments

Point 1: The only comments I want to referre are some extra proposals related to other future research case studies presented other treespecies findings in different areas and results. Collecting all the results and combine them as a vital role in future tree planting strategies.

Response 1:

Thank you for your positive comments.

Future works and research prospects have been added to the article. This work is already expanding to and developing studies in other green spaces of Coimbra, allowing for a deep and detailed analysis of the thermal effect of urban green spaces of multiple sizes and spatial configuration. The methods used in Coimbra are also being tested in other regions, namely in the city of Porto and can be expanded to other climate zones. Future research will also include reflection on tree species and their individual contributions.

Round 2

Reviewer 1 Report

All the issues were clearly answered. Accept!

Minor editing of English language required

Reviewer 2 Report

-